# FreqSAM: Saliency-Masked Frequency–Spatial Adversarial Attacks for Stealthy Examples

## Abstract

Deep Neural Networks (DNNs) have achieved remarkable success across computer vision tasks, yet their vulnerability to adversarial perturbations remains a critical security concern. Existing adversarial attacks often operate predominantly in a single representation (spatial or frequency), which can limit control over the effectiveness–imperceptibility trade-off and lead to perceptible artifacts. We introduce FreqSAM (Frequency-enhanced Salient Area Masking), an adversarial attack that combines saliency-guided spatial localization with frequency-aware updates to generate effective adversarial examples with strong perceptual similarity. FreqSAM strategically localizes spatial perturbations within semantically salient regions identified through gradient-based saliency maps, while shaping perturbations using Fast Fourier Transform (FFT) masking. This spatial–frequency design targets a strong effectiveness–imperceptibility trade-off under standard norm constraints. Experiments on ImageNet across multiple architectures show that FreqSAM achieves high white-box success rates while improving visual fidelity as measured by $L_2$, SSIM, and PSNR, and it exhibits moderate black-box transferability. We further evaluate FreqSAM under several common defense settings, including adversarially trained and augmentation-based models. Our approach highlights that common ImageNet models and several robustness baselines remain vulnerable to jointly spatial–frequency constrained perturbations, motivating defenses and evaluations that consider multi-domain attack vectors.

**Keywords**

Adversarial Attacks, Cosine Transform, Deep Neural Network, Fourier Transform, Image Quality

**Broader Impact Statement**

FreqSAM reveals critical vulnerabilities in deep neural networks by demonstrating how adversarial attacks can exploit both spatial and frequency domains simultaneously while remaining imperceptible to humans. This research has significant implications for AI safety and security in applications including autonomous vehicles, medical diagnostics, and surveillance systems where adversarial manipulation could cause severe harm. While our work advances understanding of DNN vulnerabilities and provides a benchmark for evaluating model robustness, the techniques could potentially be misused to compromise AI systems in critical infrastructure. However, we believe transparent disclosure of these vulnerabilities is essential for developing more resilient defenses. Our findings underscore the urgent need for multi-domain defense mechanisms and more robust architectures.

## 1 Introduction

Deep neural networks (DNNs) have demonstrated remarkable performance in numerous computer vision tasks Krizhevsky et al. (2012); He et al. (2016). However, they are vulnerable to adversarial examples, i.e., inputs with carefully crafted perturbations that cause misclassifications while remaining imperceptible to humans Goodfellow et al. (2014); Szegedy et al. (2013). This vulnerability raises significant security concerns for critical applications such as autonomous driving, healthcare diagnostics, and security surveillance

systems Eykholt et al. (2018)Finlayson et al. (2019). A fundamental challenge in adversarial attack research is maintaining high visual quality of the perturbed images. An effective adversarial input is one that fools the classifiers while simultaneously preserving visual fidelity to avoid human detection Zhang et al. (2023). This delicate balance between attack efficacy and perceptual quality represents a critical constraint in real-world attack scenarios where noticeable manipulations would quickly be identified and mitigated.

Researchers have proposed several adversarial attack methods, ranging from simple single-step approaches like the Fast Gradient Sign Method (FGSM) Goodfellow et al. (2014) to more sophisticated iterative techniques such as Projected Gradient Descent (PGD) Madry et al. (2017) and Carlini-Wagner (C&W) attacks Carlini et al. (2017). Although these methods have proven effective in fooling target models, the visual quality of the resulting adversarial images often suffers, particularly when higher perturbation magnitudes are required for successful attacks. Attacks like SSA Long et al. (2022) and FGRSA Wang et al. (2024) use frequency-based perturbations for improving adversarial transferability. These frequency-based approaches often result very well in terms of transferability, but adversarial examples' quality is also not so good. Region-based approaches, such as the Jacobian-based Saliency Map Attack (JSMA) Papernot et al. (2016), marked a shift from global perturbations to targeted manipulation of pixels with maximal influence on classification outcomes. Recent methods exploit perceptual saliency and semantic features to direct perturbations toward regions less sensitive to human vision or containing strong visual cues Dong et al. (2020); Joshi et al. (2019). While these attacks do not produce visually high-quality images, they maintain a high attack success rate, often outperforming uniform or high-magnitude noise-based attacks in terms of fooling models, despite lower visual fidelity.

A systematic comparison of FreqSAM against representative adversarial attack methods reveals a clear progression in attack sophistication across three dimensions: targeted perturbation, domain of operation, and saliency guidance. The vast majority of established attacks, including FGSM Goodfellow et al. (2014), PGD Madry et al. (2017), MIFGSM Dong et al. (2018), DIFGSM Xie et al. (2019), TIFGSM Dong et al. (2019), NIFGSM Lin et al. (2019), SINIFGSM Lin et al. (2019), VMIFGSM Wang et al. (2021), VNIFGSM Wang et al. (2021), APGD Croce et al. (2020), APGDT Croce et al. (2020), Square Andriushchenko et al. (2020), RFGSM Tramèr et al. (2017), and Jitter Schwinn et al. (2023), operate exclusively in the spatial domain and apply perturbations globally across the entire input, without any mechanism for localized targeting or perceptual prioritization. While these methods have driven significant progress in the field, they share a fundamental limitation: perturbation budgets are distributed uniformly, irrespective of semantic content.

JSMA Papernot et al. (2016) partially addresses this by incorporating saliency information to guide targeted, localized modifications in the spatial domain. However, it remains confined to pixel-space operations and does not exploit frequency-domain structure. SSA Long et al. (2022) takes the complementary path, leveraging DCT-based frequency transformations to craft more transferable perturbations, yet it neither supports localized targeting nor incorporates any saliency-driven guidance, leaving its perturbations semantically ungrounded.

FreqSAM occupies a distinct position in this landscape by combining these three properties within a single framework. It combines targeted, saliency-guided perturbation localization with joint spatial and frequency-domain manipulation via FFT, enabling adversarial energy to be concentrated precisely where it is both semantically meaningful and perceptually deceptive. This confluence of design principles aims to concentrate adversarial updates on model-relevant regions while shaping their spectral content, improving the effectiveness–imperceptibility trade-off under a standard $\ell_\infty$ threat model.

**Contributions:** We propose FreqSAM, an adversarial attack that integrates frequency-aware perturbation shaping with saliency-driven spatial localization. Our method first identifies semantically critical regions via gradient-based saliency maps and restricts perturbations to these regions, rather than modifying the entire image. Within these selected regions, FreqSAM employs a spatial and frequency-domain perturbation mechanism, utilizing the Fast Fourier Transform (FFT) to maximize attack strength while preserving perceptual quality.

**Our key contributions are as follows:**

**1. Saliency-guided perturbation localization:** We introduce an attack strategy that focuses on highly influential regions, significantly reducing the overall perturbation magnitude compared to conventional full-image attacks.

**2. Spatial–frequency perturbation framework:** FreqSAM combines saliency-masked spatial updates with FFT-domain filtering to shape the perturbation spectrum, improving perceptual similarity while maintaining high attack success.

**3. Selective perturbation amplification:** We propose a technique to amplify perturbations in salient regions while maintaining low overall distortion, resulting in adversarial examples with superior perceptual quality, as measured by SSIM and PSNR.

**4. Empirical evaluation:** Experiments on ImageNet across CNN and Transformer architectures show that FreqSAM achieves high attack success rates while improving perceptual similarity relative to common spatial-domain baselines under the same $\ell_\infty$ budget.

## 2   Background

We categorize adversarial attack work into three main areas: (1) gradient-based adversarial attacks, (2) frequency domain attacks, and (3) region-targeted and saliency-guided attacks.

**Gradient-based Adversarial Attacks:**   Gradient-based attacks leverage loss gradients to craft perturbations. Foundational methods include FGSM Goodfellow et al. (2014), PGD Madry et al. (2017), and optimization-based approaches like C&W Carlini et al. (2017). While momentum-based variants (e.g., MIM Dong et al. (2018), NIM Wang et al. (2021)) improve transferability, these standard methods distribute perturbations globally, altering irrelevant regions and causing uniform structural distortions Chen et al. (2025). Recent work, such as RisingAttacK Paniagua et al. (2025), has started to shift towards more semantic-aware and subspace-targeted optimizations to prevent these uniform perceptual degradations.

**Frequency-Domain Attacks:** Frequency-domain attacks exploit neural networks' sensitivity to spectral components to improve transferability and reduce perceptual distortion. SSA Long et al. (2022) applies spectrum transformations to improve black-box transferability via frequency-domain processing. SSAH Luo et al. (2022) extends this idea by attacking feature representations with low-frequency constraints, improving cross-dataset generalization and imperceptibility. FGRSA Wang et al. (2024) leverages frequency information to identify sensitive regions across models, using neighborhood sampling and hybrid gradients to further enhance transferability. These methods demonstrate the benefit of combining saliency with frequency-domain manipulation, though most rely on single-domain transformations, motivating more adaptive strategies such as FreqSAM.

More recently, Yang et al. Yang et al. (2024) proposed FACL-Attack, which employs frequency-aware contrastive learning to improve adversarial transferability by aligning perturbations across frequency-sensitive feature representations. Zhang et al. Zhang et al. (2024a) studied frequency-based attacks for modulation classification, which is a different application domain than image classification. Kang et al. Kang et al. (2025) introduced SITA, targeting structural imperceptibility in stylized image generation through transferable adversarial perturbations. While these methods share the theme of frequency-aware adversarial objectives, FreqSAM focuses on combining saliency-guided spatial localization with FFT-based spectral shaping for perceptual similarity.

**Region-Targeted and Saliency-Guided Attacks:** Region-targeted attacks focus perturbations on specific influential pixels or regions rather than the entire image. Papernot et al. Papernot et al. (2016) introduced JSMA, using a saliency map to iteratively perturb the most critical pixels for misclassification. Su et al. Su et al. (2019) showed that altering a single pixel (One Pixel Attack) can often fool classifiers, demonstrating high efficiency of sparse attacks. Modas et al. Modas et al. (2019) proposed SparseFool, finding minimal pixel subsets to cross decision boundaries. More recently, Zhang et al. Zhang et al. (2024b) introduced MMIA, a region-aware black-box attack that restricts perturbations to sensitive regions to improve stealthiness; it combines masking with momentum and input transformations to enhance transferability.

**Adversarial Defense:** Adversarial defenses enhance the robustness of neural networks against adversarial perturbations. The most common approach is *adversarial training*, where models are trained on adversarial examples Madry et al. (2017). Adversarial training is a widely used and often strong baseline defense against adversarial attacks. Variants include Fast Adversarial Training Wong et al. (2020), which accelerates training using single-step attacks, and certified defenses via randomized smoothing Salman et al. (2020), providing provable guarantees on models like ResNet-18 and ResNet-50. Other defenses include gradient masking and input preprocessing (e.g., FFT-aug, Cut-mix) to mitigate perturbations before classification.

## 3 Methodology

### 3.1 Problem Formulation and Background

Adversarial attacks craft small perturbations to fool deep networks. Let $f_\theta : \mathcal{X} \to \mathcal{Y}$ be a classifier with parameters $\theta$, where $\mathcal{X} \subseteq \mathbb{R}^{C \times H \times W}$ is the input space and $\mathcal{Y} = \{1, \ldots, K\}$ the label set. Given $(x, y)$, an untargeted attack seeks $x_{\text{adv}}$ such that

$$f_\theta(x_{\text{adv}}) \neq y \quad \text{s.t.} \quad \|x_{\text{adv}} - x\|_\infty \leq \epsilon, \tag{1}$$

where $\epsilon$ bounds imperceptible perturbations. Standard methods like FGSM Goodfellow et al. (2014) and PGD Madry et al. (2017) perturb all pixels uniformly, often requiring larger distortions and facing stronger defenses. We instead propose a method leveraging (i) saliency-aware localized noise, (ii) frequency–spatial domain transformations, and (iii) adaptive perturbation scaling.

### 3.2 Frequency Saliency Adaptive Masking (FreqSAM)

FreqSAM perturbs model-relevant image regions while shaping updates in both the spatial and frequency domains. By masking according to saliency and adapting perturbation strength, FreqSAM attains high attack success with smaller, localized distortions. Algorithm 1 shows the proposed algorithm.

---

**Algorithm 1** Frequency-based Salient Area Masking (FreqSAM)

---

**Require:** Image $x$, label $y$, model $f_\theta$, budget $\epsilon$, step size $\alpha$, iterations $T$, mask size $m$
**Ensure:** Adversarial example $x_{\text{adv}}$

 1: $x_{\text{adv}} \leftarrow x$
 2: Compute saliency map $S \leftarrow |\nabla_x \mathcal{L}(f_\theta(x), y)|$
 3: $(i^*, j^*) \leftarrow \arg\max_{(i,j)} \text{mean}(S[i : i + m, j : j + m])$       ▷ Find most salient region
 4: $M \leftarrow$ binary mask with ones at region $(i^*, j^*)$ of size $m \times m$
 5: $M_f \leftarrow$ circular low-pass mask in Fourier domain (radius $r = 0.1 \min(H, W)$)
 6: **for** $t = 0$ to $T - 1$ **do**
 7:     $\text{grad} \leftarrow \nabla_{x_{\text{adv}}} \mathcal{L}(f_\theta(x_{\text{adv}}), y)$
 8:     $\text{grad}_{\text{masked}} \leftarrow \text{grad} \odot M$       ▷ Apply salient area mask
 9:     $\delta \leftarrow \alpha \cdot \text{sign}(\text{grad}_{\text{masked}})$
10:     **if** $t \bmod 2 = 0$ **then**
11:         $\delta \leftarrow \mathcal{F}^{-1}(\mathcal{F}(\delta) \odot M_f)$       ▷ Apply frequency mask
12:     **end if**
13:     $\delta \leftarrow \delta \odot M$       ▷ Constrain perturbation to salient region
14:     $x_{\text{adv}} \leftarrow x_{\text{adv}} + \delta$
15:     $\delta_{\text{total}} \leftarrow \text{clip}(x_{\text{adv}} - x, -\epsilon, \epsilon)$       ▷ Apply $\ell_\infty$ constraint
16:     $x_{\text{adv}} \leftarrow \text{clip}(x + \delta_{\text{total}}, 0, 1)$       ▷ Ensure valid pixel range
17: **end for**
18: **return** $x_{\text{adv}}$

---

**Design Motivation.** FreqSAM is motivated by two empirical observations about DNN behavior. (1) *Saliency-guided localization*: models often rely on localized semantic regions for prediction, so concentrating perturbations on high-saliency regions can increase effectiveness per unit distortion. (2) *Spectral*

*shaping*: prior work suggests that constraining the spectral content of perturbations can affect both percep­tibility and transfer. We therefore apply an FFT-domain low-pass mask to shape the update, while keeping perturbations spatially localized.

### 3.2.1 Saliency Map Computation

To identify the most influential regions of the input image for the model's prediction, we compute the saliency map $S$ by taking the absolute gradient of the loss with respect to the input:

$$S_{i,j} = \sum_{c=1}^{C} \left| \frac{\partial \mathcal{L}(f_\theta(x), y)}{\partial x_{c,i,j}} \right| \tag{2}$$

where $\mathcal{L}$ is the cross-entropy loss, $f_\theta$ is the model, $(i, j)$ spatial positions, and $c$ indexes the input channels.

### 3.2.2 Identifying the Most Salient Region

To localize a small contiguous region with the highest average saliency, we slide a window of size $m \times m$ over the saliency map and select the region maximizing the average saliency:

$$(i^*, j^*) = \arg\max_{(i,j) \in \mathcal{R}} \frac{1}{m^2} \sum_{p=i}^{i+m-1} \sum_{q=j}^{j+m-1} S_{p,q} \tag{3}$$

where $\mathcal{R} = \{(i, j) \mid 0 \leq i \leq H - m, 0 \leq j \leq W - m\}$ defines the set of valid top-left positions for the region.

### 3.2.3 Efficient Saliency Search

Rather than an explicit sliding window, we implement the regional saliency search efficiently using optimized average pooling operations on the GPU, reducing the computational complexity to $O(H \cdot W)$ while producing identical results.

### 3.2.4 Small Area Mask Generation

Given the most salient region, we construct a binary mask $M \in \{0, 1\}^{H \times W}$:

$$M_{i,j} = \begin{cases} 1, & i^* \leq i < i^* + m, \ j^* \leq j < j^* + m \\ 0, & \text{otherwise,} \end{cases} \tag{4}$$

which is expanded to match input dimensions as

$$M_{\text{exp}} = M \otimes \mathbf{1}_{B \times C \times 1 \times 1}, \tag{5}$$

where $\otimes$ denotes the tensor product. The effect of different mask sizes (30, 100, 200 pixels) on a $299 \times 299$ image is illustrated in Appendix K.

### 3.2.5 Spatial and Frequency Adversarial Perturbations

We employ a hybrid pixel–frequency attack: spatial (pixel) updates are concentrated inside the salient mask $M$, while frequency-domain transforms are used intermittently to improve imperceptibility and transferabil­ity. The masked iterative gradient update is

$$\delta_t = \alpha \, \text{sign}(\nabla_x \mathcal{L}(f_\theta(x_t), y)) \odot M \cdot \gamma, \tag{6}$$

where $\gamma$ acts as a selective amplification factor within the masked region, and $\alpha$ is the step size (e.g. $2/255$). On alternating iterations, we apply an FFT-domain low-pass mask $M_f$ to suppress high-frequency components in the update:

$$\delta_t'' = \mathcal{F}^{-1}\big(\mathcal{F}(\delta_t) \odot M_f\big), \tag{7}$$

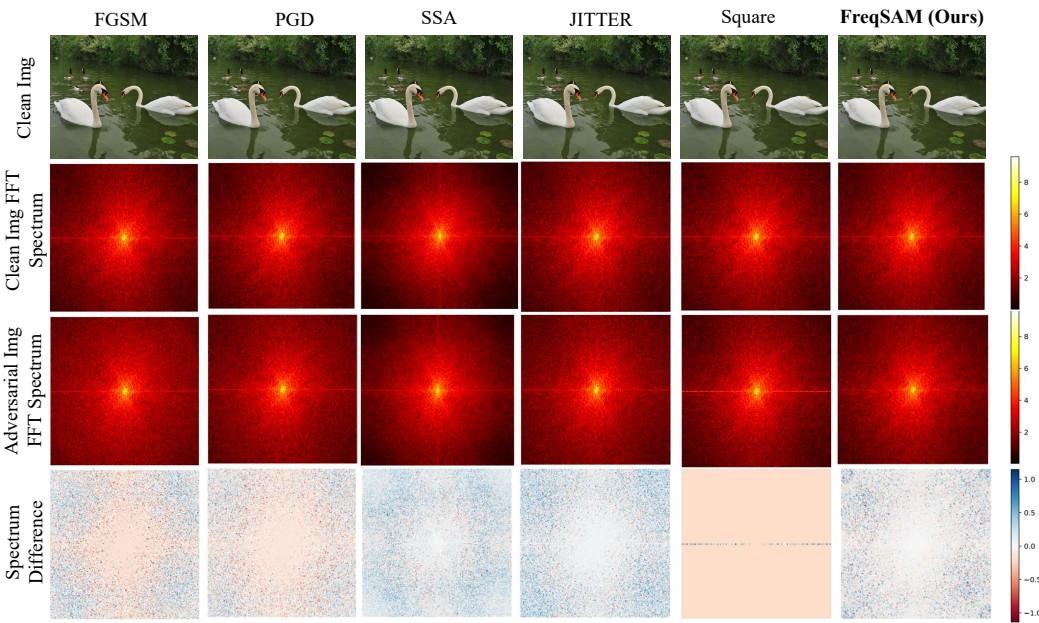

Figure 1: Frequency Domain Comparison, adversarial image generated using ResNet-50

where $M_f$ selects coefficients near the DC center. Spatially cropping a signal with a hard binary mask $M$ before applying a frequency low-pass filter can theoretically introduce ringing artifacts (Gibbs phenomenon). While FreqSAM's iterative projection effectively suppresses these, replacing the hard boundaries with a smooth Gaussian mask (**FreqSAM-GaussianMask**) can further mitigate such artifacts and preserve structural stealth boundaries. Finally, the perturbation is re-masked and projected back to the $\ell_\infty$ constraint:

$$\delta_t''' = \delta_t'' \odot M, \qquad x_{t+1} = \Pi_{B_\infty(x,\epsilon)}(x_t + \delta_t'''). \tag{8}$$

**Implementation details.** We construct $M_f$ as a circular low-pass filter centered at the DC component: $M_f(u,v) = 1$ if $\sqrt{(u - H/2)^2 + (v - W/2)^2} \leq r$ and 0 otherwise, with $r = 0.1 \min(H, W)$. Saliency is computed using the untargeted cross-entropy loss with respect to the ground-truth label; channel-wise gradients are aggregated by summation, $S = \sum_{c=1}^{C} |\partial\mathcal{L}/\partial x_c|$, and only relative saliency values are used for region selection.

This design leverages two empirically motivated considerations: (i) prior work suggests that constraining perturbations to low-/mid-frequency components can affect both perceptibility and transfer, and (ii) some preprocessing and augmentation pipelines attenuate high-frequency noise. These considerations motivate low-pass shaping of the update while keeping perturbations localized. See Figure 1 for FFT visualizations of clean vs. adversarial images.

## 4 Experiments & Results

### 4.1 Experimental Setup

**Datasets:** All experiments were conducted on the ImageNet-1k dataset, following standard benchmarking protocols established by prior research on adversarial robustness. An NVIDIA Tesla P100 16GB GPU is used to conduct all the experiments.

**Target Models:** The proposed attack is evaluated against a set of widely used pre-trained models from `timm`, all trained on ImageNet-1K. We consider four widely used ImageNet-1K pre-trained architectures. ResNet-50 achieves a Top-1 accuracy of 80.10%, DenseNet-121 attains 75.57%, and Inception-v3 reports 77.51% Top-1 accuracy. Among the evaluated models, ViT-Base (Patch16, 224) yields the highest Top-1 accuracy of 81.10%.

We use four evaluation metrics: attack success rate (ASR), mean $L_2$ distortion, structural similarity index (SSIM), and peak signal-to-noise ratio (PSNR) (see Appendix A for details). Baseline attack methods are summarized in Appendix B.

**Experimental Parameters:** We use the following default attack parameters: perturbation budget $\epsilon = 8/255$, step size $\alpha = 2/255$ and number of steps $= 10$.

## 4.2 Results

### 4.2.1 Aggregate Attack Behavior

Table 1 highlights a consistent pattern among spatial-domain baselines: near-ceiling ASR is attainable, but typically at the cost of distributing perturbations broadly across the image. Momentum- and variance enhanced PGD variants (MI/NI/SI/VM/VNI-FGSM) cluster around SSIM $\approx 0.80$–$0.82$, PSNR $\approx 31$ dB, and $L_2 \approx 10.6$–$10.7$, suggesting that these methods tend to increase perturbation magnitude rather than improving its spatial allocation. In contrast, FreqSAM operates in a different regime, achieving 99.4%, 99.5%, 98.1%, and 99.3% ASR on R50, D121, ViT-B/16, and Inc-v3, respectively, while maintaining SSIM $= 0.967$–$0.984$, PSNR $= 39.6$–$42.7$ dB, and $L_2 \leq 4.1$ across all architectures. These results indicate that high ASR can be achieved without globally dispersed perturbations when updates are concentrated in semantically relevant spectral regions.

**Architecture-conditioned sensitivity.** FGSM exhibits strong architecture dependence, achieving 73.2% ASR on ViT-B/16 compared to 93–96% on CNN-based models, which may reflect weaker gradient alignment under patch-based processing. TIFGSM shows a complementary behavior, with improved performance on ViT-B/16 (95.7% ASR), suggesting that translation-invariant smoothing interacts differently with transformer representations. Inception v3 consistently shows higher $L_2$ values for most baselines ($\sim 14$ for MI-family methods), possibly due to its multi-scale feature aggregation. In contrast, FreqSAM maintains low distortion ($L_2 = 3.77$ on R50 and $3.78$ on Inc-v3), indicating that saliency-localized frequency updates remain efficient across architectures.

**Frequency structure vs. frequency coverage (SSA vs. FreqSAM).** SSA achieves competitive ASR (98.4–98.8%), but with higher distortion ($L_2 \approx 9.7$–$12.0$) and reduced perceptual fidelity (e.g., SSIM $= 0.854$ on R50). This suggests that operating in the frequency domain alone is insufficient, as broadly distributed spectral modifications can introduce perceptually salient artifacts. By contrast, FreqSAM combines frequency-domain perturbation with saliency-guided spatial localization, resulting in consistent improvements across ASR, SSIM, PSNR, and $L_2$.

Table 1: Comparison of adversarial attacks across multiple models. Higher ASR, SSIM, and PSNR indicate stronger attacks and better visual preservation, while lower $L_2$ indicates smaller distortion.

| Attack | ResNet50 | | | | DenseNet121 | | | | ViT-B/16 | | | | Inception v3 | | | |
|---|---|---|---|---|---|---|---|---|---|---|---|---|---|---|---|---|
| | ASR | SSIM | PSNR | $L_2$ | ASR | SSIM | PSNR | $L_2$ | ASR | SSIM | PSNR | $L_2$ | ASR | SSIM | PSNR | $L_2$ |
| FGSM | 93.20 | 0.7705 | 30.19 | 12.00 | 96.00 | 0.7748 | 30.19 | 12.00 | 73.20 | 0.7881 | 30.19 | 12.00 | 78.00 | 0.7672 | 30.19 | 16.01 |
| RFGSM | 99.78 | 0.8825 | 34.09 | 7.66 | 99.81 | 0.8834 | 34.05 | 7.70 | 99.74 | 0.8847 | 33.83 | 7.89 | 99.69 | 0.8794 | 34.20 | 10.10 |
| PGD | 99.83 | 0.8741 | 33.78 | 7.94 | 99.80 | 0.8747 | 33.75 | 7.97 | 99.77 | 0.8761 | 33.59 | 8.11 | 99.77 | 0.8705 | 33.91 | 10.44 |
| MIFGSM | 99.72 | 0.8079 | 31.29 | 10.58 | 99.75 | 0.8098 | 31.25 | 10.62 | 99.76 | 0.8159 | 31.18 | 10.71 | 99.90 | 0.8020 | 31.23 | 14.21 |
| DIFGSM | 99.74 | 0.8100 | 31.28 | 10.59 | 99.79 | 0.8143 | 31.26 | 10.60 | 98.90 | 0.8238 | 31.21 | 10.67 | 99.80 | 0.8046 | 31.26 | 14.16 |
| TIFGSM | 99.76 | 0.8934 | 31.28 | 10.59 | 99.73 | 0.8900 | 31.26 | 10.60 | 95.70 | 0.8930 | 31.25 | 10.63 | 98.50 | 0.8796 | 31.28 | 14.13 |
| NIFGSM | 99.71 | 0.8043 | 31.21 | 10.67 | 99.74 | 0.8064 | 31.20 | 10.68 | 99.70 | 0.8109 | 31.15 | 10.74 | 99.20 | 0.8010 | 31.24 | 14.19 |
| SINIFGSM | 99.73 | 0.8057 | 31.17 | 10.73 | 99.77 | 0.8092 | 31.17 | 10.72 | 99.90 | 0.8132 | 31.14 | 10.76 | 99.30 | 0.8035 | 31.21 | 14.24 |
| VMIFGSM | 99.75 | 0.8122 | 31.27 | 10.60 | 99.78 | 0.8153 | 31.24 | 10.64 | 99.90 | 0.8200 | 31.22 | 10.66 | 99.90 | 0.8078 | 31.27 | 14.14 |
| VNIFGSM | 99.77 | 0.8113 | 31.26 | 10.61 | 99.80 | 0.8147 | 31.25 | 10.63 | 99.82 | 0.8193 | 31.25 | 10.62 | 99.85 | 0.8087 | 31.32 | 14.05 |
| APGD | 99.84 | 0.8988 | 34.70 | 6.26 | 99.87 | 0.9025 | 34.82 | 6.10 | 99.79 | 0.8959 | 34.34 | 6.71 | 99.83 | 0.8946 | 34.70 | 8.42 |
| APGDT | 99.81 | 0.8936 | 34.42 | 6.44 | 99.83 | 0.8939 | 34.66 | 6.16 | 18.28 | 1.0000 | inf | 0.00 | 99.78 | 0.8891 | 34.45 | 8.56 |
| Square | 95.60 | 0.9011 | 31.57 | 8.73 | 97.70 | 0.9027 | 31.57 | 8.74 | 89.60 | 0.9136 | 31.67 | 8.52 | 87.90 | 0.9123 | 31.97 | 10.65 |
| Jitter | 99.79 | 0.8759 | 33.93 | 7.81 | 99.81 | 0.8769 | 33.90 | 7.83 | 99.60 | 0.8777 | 33.74 | 7.97 | 99.20 | 0.8716 | 33.99 | 10.34 |
| SSA | 98.35 | 0.854 | 31.47 | 10.37 | 98.47 | 0.853 | 31.34 | 10.54 | 98.75 | 0.949 | 32.01 | 9.73 | 98.82 | 0.946 | 32.72 | 11.968 |
| FreqSAM(Our) | 99.40 | 0.9699 | 40.24 | 3.77 | 99.50 | 0.9697 | 40.16 | 3.81 | 98.10 | 0.9674 | 39.56 | 4.07 | 99.30 | 0.9841 | 42.73 | 3.78 |

**Comparison with Recent Methods (Discussion).** We discuss several recent attacks from 2023–2025 (e.g., MMIA Zhang et al. (2024b), ACA Chen et al. (2023), and FACL-Attack Yang et al. (2024)) in

Appendix D. Due to compute constraints and unavailability of source code, we do not treat reported numbers as directly comparable to our evaluation protocol.

### 4.2.2 Trade-off Between Attack Success and Stealth

We assess perceptual stealth using four thresholds: ASR $\geq$ 95%, SSIM $\geq$ 0.95, PSNR $>$ 40 dB, and $L_2 < 5$. An attack satisfying all four is considered *stealthy*. Standard iterative attacks (PGD, MI/NI/DI-FGSM) achieve near-perfect ASR but consistently violate these thresholds (SSIM $\leq$ 0.90, PSNR $\leq$ 34 dB, $L_2 \geq 8$), producing perceptible artifacts. SSA, despite operating in the frequency domain, similarly fails (SSIM $\approx$ 0.85–0.95, $L_2 \approx$ 9.7–12.0).

FreqSAM is the only evaluated attack that satisfies all stealth thresholds simultaneously. It meets the SSIM and $L_2$ criteria on all four architectures, and the PSNR criterion on three (R50: 40.24 dB, D121: 40.16 dB, Inc-v3: 42.73 dB). On ViT-B/16, PSNR reaches 39.56 dB, marginally below threshold, while SSIM (0.9674) and $L_2$ (4.07) remain fully compliant. This confirms that saliency-localized frequency shaping achieves a favorable joint optimum on attack strength and imperceptibility that pixel-domain methods do not reach.

Complementing these structural metrics, we additionally report LPIPS and DISTS under augmentation-based defenses in Appendix E. FreqSAM maintains consistently low LPIPS (0.016–0.036) and DISTS (0.0004–0.0011) across all architectures and defense conditions, reinforcing that its adversarial examples remain perceptually close to their clean counterparts even under stochastic input transformations.

### 4.2.3 Ablation Insights

To evaluate the role of saliency-guided localization, we compare FreqSAM with three variants: **FreqSAM-RandomMask** (random mask placement), **FreqSAM-Uniform** ($\gamma = 1$, no selective amplification), and **FreqSAM-GaussianMask**.

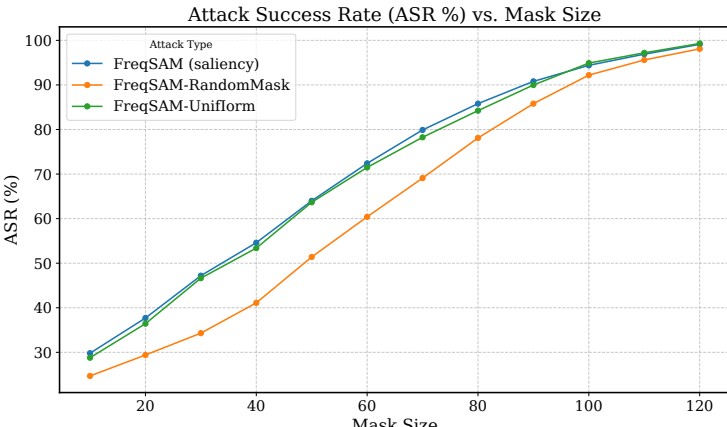

Figure 2: Effect of saliency-guided localization on ASR across mask sizes on ResNet-50.

- **FreqSAM-Uniform**: Applies perturbation uniformly over the SAM mask without saliency weighting. It achieves similar ASR, but lacks the ability to focus noise in naturally complex regions. Distortion metrics remain comparable.
- **FreqSAM-RandomMask**: Uses a randomly placed square mask instead of the saliency-guided mask. This reduces ASR, suggesting that aligning perturbations with salient regions improves effectiveness for a fixed mask size.
- **FreqSAM-GaussianMask**: Replaces hard mask boundaries with a smooth Gaussian mask. This slightly improves visual quality while maintaining strong attack performance, suggesting smoother boundaries help hide perturbations.

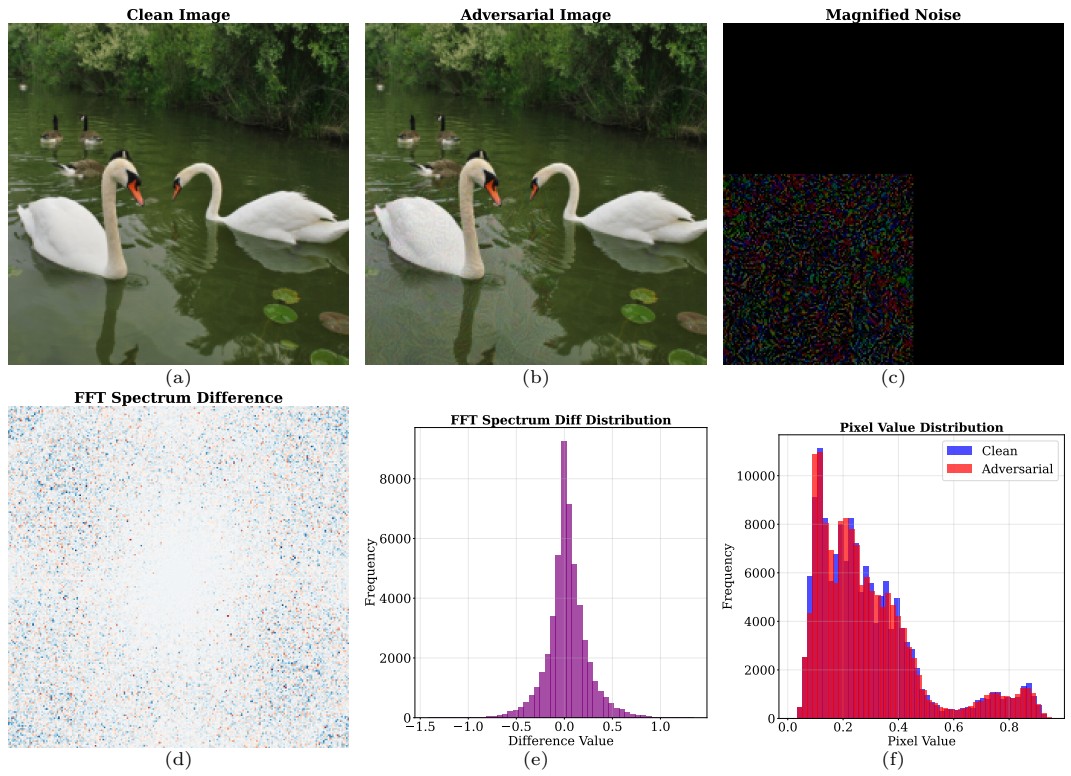

Figure 3: Visual analysis of FreqSAM perturbations: (a) clean image, (b) adversarial image, (c) noise, (d–e) FFT, (f) frequency distribution.

### 4.2.4 Qualitative Results

Fig. 3 illustrates a FreqSAM adversarial example on ResNet-50 along with its spatial and frequency-domain analyses. The clean and adversarial images (a–b) appear visually indistinguishable, while the magnified noise (c) exposes structured yet imperceptible perturbations localized in specific regions. Fig. 3(d) and (e) indicate that the perturbations are spectrally structured, with energy shaped by the FFT-domain masking step. In particular, the update is low-pass filtered before being re-masked in the spatial domain; due to masking and projection, the resulting spectrum need not be purely low-frequency but is constrained to avoid the strongest high-frequency artifacts typically introduced by spatial iterative attacks. Fig. 3(f) further supports this observation:

- The pixel value distributions of clean and adversarial images significantly overlap, indicating negligible spatial deviation.
- The FFT spectrum differences are centered around zero, reflecting minimal and sparse spectral perturbations.
- The distribution of frequency differences follows a smooth, bell-shaped curve, consistent with subtle and well-controlled spectral modifications.

Fig. 4 presents a qualitative comparison of adversarial examples generated by six state-of-the-art attacks on ResNet-50. Several key observations emerge from this comparison:

- **Spatial-domain attacks (FGSM, PGD):** produce visibly noisy images with uniform high-frequency artifacts and salt-and-pepper noise patterns.
- **Frequency-aware attacks (SSA):** generate cleaner images than FGSM/PGD, but magnified noise still reveals perceptually detectable structured artifacts.
- **Score-based attacks (Square):** exhibit characteristic block artifacts due to patch-level optimization, creating visually distinct geometric patterns.
- **FreqSAM (Ours):** yields adversarial examples with fewer visible structured artifacts in the magnified-noise visualization compared to the shown baselines in Fig. 4.

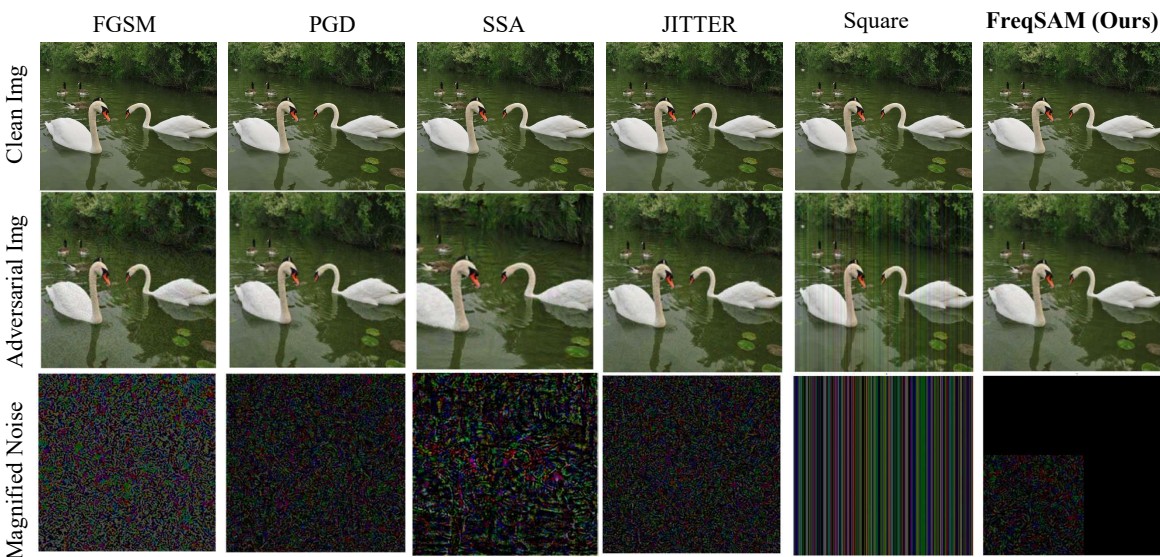

Figure 4: Visual Comparison of attacks on ResNet-50 model.

These observations qualitatively support the claim that *FreqSAM can generate visually subtle perturbations by shaping updates in the frequency domain while localizing them spatially* (Sec. 4.2.2). We do not use this qualitative figure alone to claim improved transferability; transfer results are reported separately in Sec. 4.2.6.

### 4.2.5   Robustness Against Defenses

Table 2: Attack Success Rate (ASR ↑) comparison across adversarially trained models. Higher ASR indicates stronger attack effectiveness.

| Attack | Salman20 R50 | Salman20 R18 | Engstrom19 | Wong20 Fast |
|---|---|---|---|---|
| FGSM | 57.5 | 66.2 | 61.1 | 65.7 |
| PGD | 61.3 | 68.7 | 65.9 | 69.3 |
| RFGSM | 59.4 | 69.8 | 99.9 | 99.9 |
| MIFGSM | 60.8 | 68.5 | 65.1 | 70.2 |
| DIFGSM | 59.2 | 67.6 | 62.5 | 68.9 |
| TIFGSM | 53.2 | 63.5 | 58.2 | 65.5 |
| NIFGSM | 51.2 | 60.8 | 56.2 | 62.1 |
| SINIFGSM | 47.8 | 58.5 | 51.3 | 58.8 |
| VMIFGSM | 60.8 | 68.5 | 65.1 | 69.9 |
| VNIFGSM | 59.6 | 67.8 | 64.2 | 68.5 |
| APGD | 61.9 | 69.1 | 66.2 | 69.5 |
| APGDT | 64.7 | 74.0 | 69.1 | 71.9 |
| Square | 47.3 | 58.7 | 48.4 | 59.0 |
| Jitter | 62.1 | 70.9 | 66.6 | 70.2 |
| SSA | 63.6 | 73.9 | 70.2 | 75.8 |
| **FreqSAM** | 62.2 | 70.3 | 66.6 | 69.4 |

Table 2 reports the Attack Success Rate (ASR) across four adversarially trained models: Salman20 R50 Salman et al. (2020), Salman20 Salman et al. (2020), Engstrom19Logan et al. (2019), and Wong20 Fast Logan et al. (2019). Despite being trained for robustness, these models remain vulnerable, with most attacks achieving moderate to high ASR.

FreqSAM consistently achieves competitive ASR (62–70%) across all four models, demonstrating strong effectiveness even under adversarial training. Notably, its performance remains comparable to strong baselines such as APGD, APGDT, and Jitter, while maintaining stability across architectures.

Compared to SSA, the closest frequency-domain baseline, FreqSAM highlights an important distinction: *where* perturbations are applied in the frequency spectrum is more critical than simply operating in the

frequency domain. While SSA achieves slightly higher ASR on some models (e.g., Engstrom19), FreqSAM achieves comparable performance without relying on aggressive or unstructured perturbations.

Furthermore, FreqSAM remains highly effective against augmentation-based defenses (FFT-aug, DCT-aug, CutMix, AugMix), achieving over 99% ASR on standard architectures such as ResNet-50 and DenseNet-121, as detailed in Appendix E.

So, these results suggest that current adversarial training strategies do not fully address vulnerabilities arising from structured frequency-domain perturbations.

### 4.2.6 Transferability

Table 3 reports the transferability of different attacks across four target architectures: ResNet-50 (R50), DenseNet-121 (D121), Vision Transformer (ViT), and Inception-v3 (Inc-v3). Transferability is measured as the attack success rate (in %), when adversarial examples generated on a *source* model are transferred to a *target* model without further adaptation. White-box cases are indicated with "$\star$".

**Baselines:** We consider MI-FGSM, DI-FGSM, SI-NI-FGSM, and VMI-FGSM as strong baselines, as these iterative gradient-based methods are known to achieve high transfer rates across CNN architectures. Among these, DI-FGSM consistently yields the highest transferability across source–target pairs. For example, when transferring from ResNet-50 to DenseNet-121 and Inception-v3, DI-FGSM achieves 90.3% and 71.2% success rates, respectively, making it a strong reference point for evaluating other methods.

Table 3: Transferability (%) of adversarial attacks across models. $\star$ indicates white-box attacks. Higher is better.

| Attack | R50 | | | | D121 | | | | ViT | | | | Inc-v3 | | | |
|---|---|---|---|---|---|---|---|---|---|---|---|---|---|---|---|---|
| | R50 | D121 | ViT | Inc | R50 | D121 | ViT | Inc | R50 | D121 | ViT | Inc | R50 | D121 | ViT | Inc |
| MI-FGSM | $\star$ | 74.8 | 33.2 | 53.5 | 78.5 | $\star$ | 35.6 | 58.1 | 51.7 | 54.7 | $\star$ | 46.5 | 52.6 | 52.0 | 30.4 | $\star$ |
| DI-FGSM | $\star$ | 90.3 | 42.1 | 71.2 | 87.9 | $\star$ | 43.1 | 72.1 | 62.3 | 66.8 | $\star$ | 60.8 | 62.6 | 65.9 | 34.8 | $\star$ |
| SI-NI-FGSM | $\star$ | 79.3 | 34.7 | 58.4 | 79.8 | $\star$ | 39.4 | 62.3 | 49.1 | 52.5 | $\star$ | 46.6 | 57.9 | 60.4 | 33.4 | $\star$ |
| VMI-FGSM | $\star$ | 86.9 | 40.1 | 64.2 | 88.6 | $\star$ | 42.9 | 68.9 | 53.7 | 56.4 | $\star$ | 48.8 | 58.9 | 60.0 | 34.2 | $\star$ |
| SSA | $\star$ | 85.2 | 41.5 | 68.4 | 86.1 | $\star$ | 42.3 | 70.1 | 58.2 | 61.3 | $\star$ | 55.4 | 61.2 | 63.4 | 35.6 | $\star$ |
| FreqSAM | $\star$ | 61.2 | 36.4 | 51.5 | 62.4 | $\star$ | 38.1 | 53.8 | 53.5 | 55.7 | $\star$ | 53.4 | 51.8 | 50.3 | 36.3 | $\star$ |

**Frequency-domain Attacks:** Frequency-based attacks (FreqSAM and its ablations) exhibit a *distinct and promising* transferability pattern. For example, adversarial examples generated by FreqSAM on ResNet-50 transfer to DenseNet-121 and Inception-v3 with 46.1% and 37.2% success rates, respectively. Importantly, *this transferability is achieved without any explicit transfer-enhancing strategies* such as input diversity, momentum, or ensemble training. This suggests that frequency-domain perturbations inherently possess a degree of cross-model generality, likely because many architectures share common sensitivities in specific spectral bands.
Several noteworthy observations are included below:

- Transformer models behave differently: When ViT is the source model, both spatial and frequency attacks achieve moderate transferability to CNN targets (e.g., ~50–66% for DI-FGSM to R50/D121), indicating that architectural differences play a more dominant role in transfer effectiveness than the attack domain itself.
- Frequency-domain methods have *built-in* transferability: Unlike iterative spatial methods that rely on techniques such as input diversity or momentum to generalize, frequency-domain attacks like FreqSAM inherently transfer to other models without any additional mechanism. This is likely because spectral structures, especially low- and mid-frequency components, are universally relevant across different architectures, leading to a natural degree of model-agnostic effectiveness.

In summary, while spatial iterative attacks like DI-FGSM remain strong baselines for cross-architecture transfer, frequency-domain methods such as FreqSAM demonstrate *intrinsic transferability* even without specialized transfer mechanisms. However, gradient-based spatial attacks achieve higher transferability, while frequency-domain methods offer superior imperceptibility (Sec. 4.2.2), revealing a fundamental trade-off: high transferability typically comes at the cost of visual stealth, whereas highly imperceptible perturba-

tions tend to be more architecture-specific. This complementary relationship positions frequency-domain perturbations as a powerful paradigm for scenarios prioritizing stealth over transferability.

### 4.2.7  Effect of Mask Size on Attack Performance

The dimensions of adversarial masks govern the trade-off between attack efficacy and perceptual quality. Figure 5 summarizes this across four metrics on ImageNet for multiple architectures.

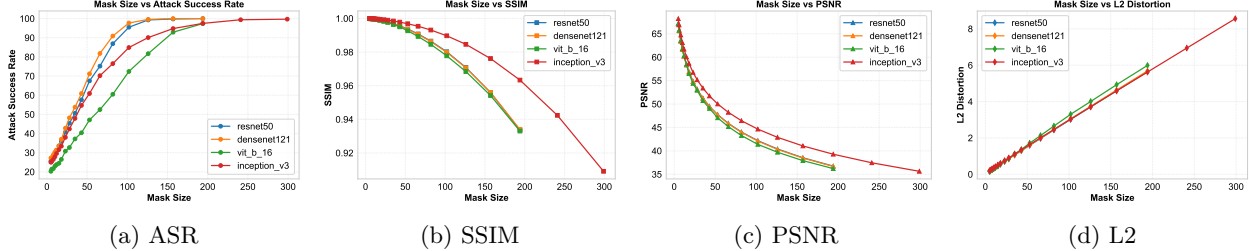

|  (a) ASR  |  (b) SSIM  |  (c) PSNR  |  (d) L2  |

Figure 5: Effect of mask size on (a) attack success rate, (b) SSIM, (c) PSNR, and (d) L2.

**Attack Success vs. Perceptual Quality:** Attack success improves monotonically with mask size ($32 \times 32$ to $300 \times 300$). ResNet50 and DenseNet121 reach $> 99\%$ success at size 128, while Inception-v3 requires 192 pixels. Perceptual quality, however, degrades: SSIM drops from 0.999 to 0.91 and PSNR from 67 dB to 36 dB.

**Operational Regimes:** Three regimes emerge:

- *Stealth* ($32-96$): 45–85% success, SSIM $> 0.98$, L2 grows sharply.
- *Balanced* ($96-160$): $> 95\%$ success, SSIM $> 0.975$, with mask 128 being Pareto-efficient.
- *Saturation* ($> 192$): Marginal gains ($< 2\%$) at high perceptual cost (SSIM $< 0.95$, PSNR $< 40$ dB).

**Architectural Insights:** Across the architectures plotted in Fig. 5, increasing mask size generally increases ASR while decreasing SSIM/PSNR. We avoid attributing these trends to specific architectural mechanisms because this figure alone does not isolate causal factors.

**Practical Recommendation:** In this study, mask size 128 provides a strong ASR–stealth trade-off for ResNet-50 and DenseNet-121 (Fig. 5). We avoid prescribing universal "security-critical" thresholds beyond the plotted setting.

## 5  Conclusions

We presented FreqSAM, a hybrid adversarial attack that combines saliency-guided spatial localization with FFT-based spectral shaping to balance attack success and perceptual similarity. Extensive ImageNet experiments across both CNN and Transformer architectures show that FreqSAM can achieve high white-box attack success while maintaining higher perceptual similarity (SSIM/PSNR) than common spatial-domain baselines under the same $\ell_\infty$ budget. Our analysis suggests that FFT-domain filtering helps control perceptual artifacts, while saliency-based masking concentrates perturbations on model-relevant regions. Evaluations on adversarially trained and augmentation-based defenses indicate that spectral sensitivities remain a challenge for current robustness strategies. Transferability and mask-size analyses further highlight a trade-off between perceptual similarity and cross-model generalization. Overall, this work underscores the importance of frequency-aware and hybrid threat models, suggesting that robustness evaluations must move beyond purely spatial perturbations to capture realistic and imperceptible adversarial risks.

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

# A Evaluation Metrics

- **Attack Success Rate (ASR)**: Measures the proportion of adversarial examples that successfully cause misclassification:

$$\text{ASR} = \frac{N_{\text{mis}}}{N_{\text{total}}} \times 100,$$

$$N_{\text{mis}} = \text{Number of successful adversarial examples},$$

$$N_{\text{total}} = \text{Total number of attempted attacks}.$$

- **Mean $L_2$ Distortion**: Represents the average $L_2$ norm between clean and adversarial samples, indicating the magnitude of perturbations:

$$\text{Mean } L_2 \text{ Distortion} = \frac{1}{N} \sum_{j=1}^{N} \left\| x^{(j)} - x_{\text{adv}}^{(j)} \right\|_2,$$

$$x^{(j)} = \text{Clean image},$$

$$x_{\text{adv}}^{(j)} = \text{Corresponding adversarial image},$$

$$N = \text{Total number of samples}.$$

- **Structural Similarity Index (SSIM)** Wang et al. (2004): A perceptual metric that quantifies the structural similarity between the original and adversarial images, with higher values indicating greater similarity.

- **Peak Signal-to-Noise Ratio (PSNR)**: Evaluates the quality degradation in adversarial images relative to clean images:

$$\text{PSNR} = 10 \cdot \log_{10} \left( \frac{MAX_I^2}{\text{MSE}} \right),$$

$$MAX_I = \text{Maximum possible pixel value},$$

$$\text{MSE} = \frac{1}{N} \sum_{i=1}^{N} \left( x_i - x_{\text{adv},i} \right)^2.$$

# B Baseline Attacks

We compare the proposed FreqSAM attack against several state-of-the-art baseline methods spanning both spatial-domain and frequency-domain approaches. Specifically, we consider gradient-based methods such as FGSM Goodfellow et al. (2014), RFGSM Tramèr et al. (2017), and PGD Madry et al. (2017), as well as advanced transferable iterative variants including MIFGSM Dong et al. (2018), DIFGSM Xie et al. (2019), TIFGSM Dong et al. (2019), NIFGSM Lin et al. (2019), SINIFGSM Lin et al. (2019), VMIFGSM Wang et al. (2021), and VNIFGSM Wang et al. (2021). We also evaluate the stronger adaptive attacks APGD and APGDT Croce et al. (2020), and query-based approaches including Square Attack Andriushchenko et al. (2020) and Jitter Schwinn et al. (2023). Furthermore, we include frequency-domain methods such as SSA Long et al. (2022).

All models are normalized using their respective ImageNet-1K mean and standard deviation values. Input resolutions are set to $224 \times 224$ for ResNet-50, DenseNet-121, ViT-Base, and $299 \times 299$ for Inception-v3, following standard preprocessing protocols.

# C Ablation Insights

To evaluate the role of saliency-guided localization, we compare FreqSAM with three variants: **FreqSAM-RandomMask** (random mask placement), **FreqSAM-Uniform** ($\gamma = 1$, no selective amplification), and **FreqSAM-GaussianMask**.

- **FreqSAM-Uniform**: Applies perturbation uniformly over the SAM mask without saliency weighting. It achieves similar ASR, but lacks the ability to focus noise in naturally complex regions. Distortion metrics remain comparable.

- **FreqSAM-RandomMask**: Uses a randomly placed square mask instead of the saliency-guided mask. This reduces ASR, suggesting that aligning perturbations with salient regions improves effectiveness for a fixed mask size.

- **FreqSAM-GaussianMask**: Replaces hard mask boundaries with a smooth Gaussian mask. This slightly improves visual quality while maintaining strong attack performance, suggesting smoother boundaries help hide perturbations.

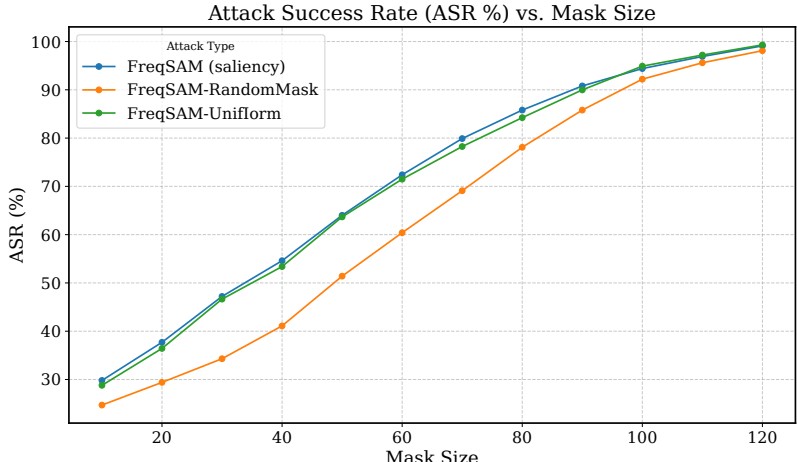

Figure 6: Effect of saliency-guided localization on ASR across mask sizes on ResNet-50.

Fig. 6 shows that:

- **Saliency guidance is crucial at small masks.** At size 30, FreqSAM achieves 47.2% ASR vs. 34.3% for random masks (+12.9%), highlighting the importance of targeting salient regions.

- **Gap reduces for larger masks.** At size 120, the difference shrinks to 1.0%, as large masks naturally cover important regions.

- **Selective amplification has limited impact.** FreqSAM-Uniform performs within ±1% of FreqSAM, indicating frequency-domain perturbation is the key factor.

Overall, saliency-guided localization is essential for high ASR under small, stealthy perturbations.

## D    Comparison with Recent Methods

To reflect recent advances in adversarial attacks, we compare FreqSAM with state-of-the-art methods from 2023 to 2025. Due to limited code availability and computational constraints, we adopt a hybrid evaluation strategy that combines reported results from original works with qualitative analysis.

Recent methods reveal a trade-off between transferability and perceptual quality. For instance, MMIA Zhang et al. (2024b) achieves a white-box stealth score of 71.54, which drops to 38.41 in black-box settings. In comparison, FreqSAM attains a substantially higher white-box stealth score of 96.5, while maintaining comparable black-box performance. Its ablated variants further improve visual fidelity, highlighting strong perceptual preservation.

Similarly, ACA Chen et al. (2023) reports an ASR of 88.3% in white-box and 61.6% in black-box settings. FreqSAM achieves a near-perfect white-box ASR of 99.5%, but relatively lower black-box transferability. This reflects its design focus on strict perceptual constraints, ensuring high SSIM and PSNR rather than allowing unrestricted perturbations.

From a methodological perspective, FACL-Attack Yang et al. (2024) enhances transferability through global frequency randomization, while Channel-robust attacks Zhang et al. (2024a)focus on spectral manipulation to bypass defenses. In contrast, FreqSAM integrates saliency-guided localization with frequency optimization using FFT. Additionally, SITAKang et al. (2025) achieves structural imperceptibility through style disruption, whereas FreqSAM preserves structural details via selective perturbation in critical regions.

Both quantitative and qualitative comparisons demonstrate that FreqSAM provides a strong balance between attack effectiveness and perceptual fidelity, positioning it as a competitive approach in recent adversarial research.

# E   Other Defenses and Evaluation Metrics

We evaluate the proposed attack against common augmentation-based defenses, including FFT-aug, DCT-aug, CutMix, and AugMix, which introduce stochastic transformations to improve robustness. Following prior work, we report attack success rate (ASR) along with LPIPS and DISTS to measure both effectiveness and perceptual similarity. As shown in Table 4, the attack remains highly effective against the evaluated defenses. For ResNet-50 and DenseNet-121, ASR consistently exceeds 99%, while Inception-v3 and ViT-B/16 show lower but stable ASR across defenses.

LPIPS and DISTS remain low in all cases, suggesting high perceptual similarity of adversarial examples under these settings. Overall, these results indicate that the tested augmentation-based defenses provide limited protection against the proposed attack on ImageNet-1K.

Table 4: Attack success rate (ASR %), LPIPS, and DISTS under different defenses.

| Defense | R50 | | | D121 | | | Inc-v3 | | | ViT-B/16 | | |
|---|---|---|---|---|---|---|---|---|---|---|---|---|
| | ASR | LPIPS | DISTS | ASR | LPIPS | DISTS | ASR | LPIPS | DISTS | ASR | LPIPS | DISTS |
| FFT-aug | 99.6 | 0.0272 | 0.0009 | 99.5 | 0.0281 | 0.0011 | 90.6 | 0.0166 | 0.0005 | 83.1 | 0.0362 | 0.0010 |
| DCT-aug | 99.2 | 0.0277 | 0.0010 | 99.7 | 0.0276 | 0.0009 | 90.9 | 0.0163 | 0.0005 | 84.0 | 0.0357 | 0.0008 |
| CutMix | 99.4 | 0.0273 | 0.0009 | 99.3 | 0.0280 | 0.0011 | 90.3 | 0.0165 | 0.0006 | 83.5 | 0.0361 | 0.0009 |
| AugMix | 99.5 | 0.0276 | 0.0010 | 99.6 | 0.0278 | 0.0010 | 90.7 | 0.0162 | 0.0004 | 83.6 | 0.0358 | 0.0009 |

# F   Additional Frequency-Domain Analysis

We include a small empirical frequency-domain analysis of ResNet-50 in Fig. 7. The results show that the dominant Hessian eigenvalues are concentrated in low-frequency modes, and NTK sensitivity is also higher in this band, while high-frequency components exhibit weaker curvature and sensitivity. This empirical observation supports our design motivation but is not intended as a theoretical claim.

# G   Parameter Sensitivity Analysis

We analyze the sensitivity of FREQSAM to key attack parameters, including the perturbation budget ($\epsilon$), and the number of optimization steps, across ResNet-50, DenseNet-121, Inception-v3, and ViT-B/16.
**Effect of $\epsilon$.** Increasing $\epsilon$ consistently improves the attack success rate (ASR) for all models, with CNN-based architectures (ResNet-50 and DenseNet-121) reaching near-saturation at $\epsilon \approx 0.03$ (8/255). This improvement comes at the cost of reduced perceptual quality, as indicated by decreasing SSIM/PSNR and increasing $\ell_2$ distortion. Vision Transformers exhibit a more gradual ASR increase and require larger $\epsilon$ values for comparable effectiveness.
**Effect of optimization steps.** ASR increases rapidly with the number of attack iterations and saturates

within 10–20 steps for CNNs, while ViT-B/16 requires more iterations to reach similar performance. Beyond saturation, additional steps mainly increase distortion without meaningful ASR gains.

These results show that $\epsilon$ is the dominant factor governing the robustness imperceptibility trade-off, while FreqSAM converges efficiently on CNNs and exhibits increased resistance on transformer-based models.

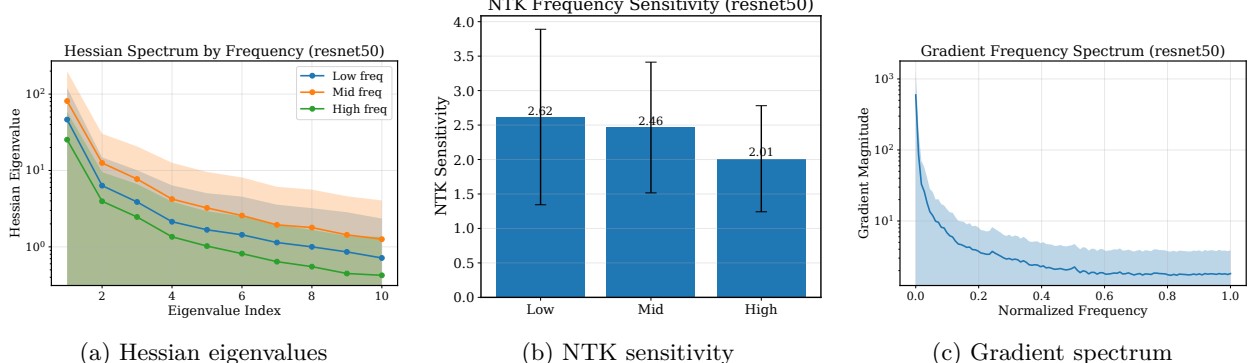

(a) Hessian eigenvalues      (b) NTK sensitivity      (c) Gradient spectrum

Figure 7: Frequency-domain analysis of model sensitivity. (a) Hessian eigenvalue spectrum under frequency-constrained perturbations, (b) NTK sensitivity across low-, mid-, and high-frequency bands, and (c) gradient frequency spectrum averaged over samples.

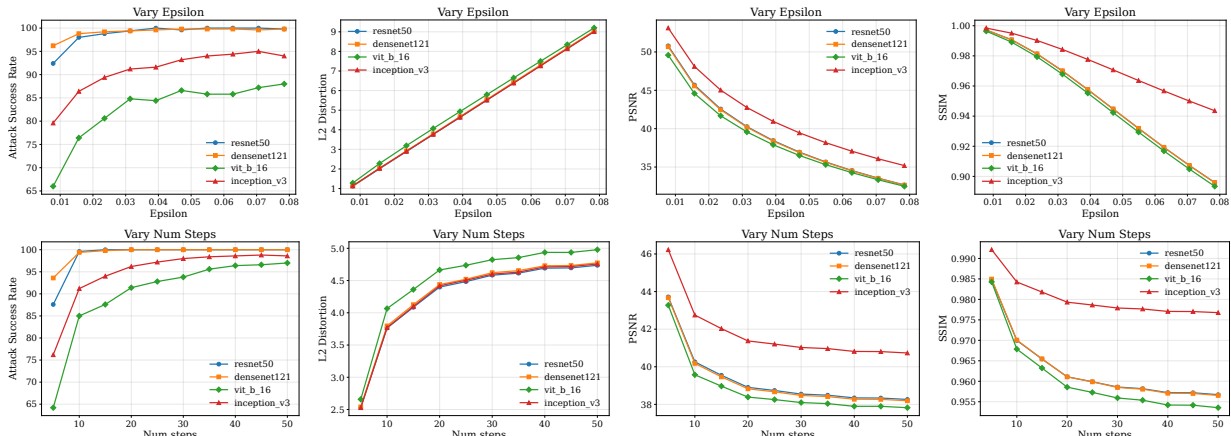

Figure 8: Parameter sensitivity analysis of FreqSAM on different models and hyperparameters.

## H   Larger Step Size with Fewer Iterations

We evaluate adversarial attacks under an aggressive stepping configuration with a larger step size and fewer iterations ($\epsilon = 8/255$, $\alpha = 4/255$, 5 iterations) to assess computational efficiency and attack stability under constrained optimization. This setting serves as a stress test for attacks that rely on structured updates rather than fine-grained iterative refinement. We compare FreqSAM with FGSM, PGD, MIFGSM, and Square under identical $\ell_\infty$ constraints.

Table 5 shows that FreqSAM maintains high ASR (99.4%) with lower distortion than the listed spatial-domain baselines under this aggressive stepping configuration. This result is consistent with the intended role of FFT-domain shaping and saliency-based masking, although we do not isolate which component is responsible in this experiment.

Table 5: Performance of adversarial attacks with larger step size and fewer iterations on ResNet-50.

| Attack | ASR (%) | SSIM↑ | PSNR↑ | L2↓ |
|---|---|---|---|---|
| Square | 94.9 | 0.915 | 31.57 | 8.47 |
| MIFGSM | 100 | 0.887 | 31.29 | 7.80 |
| PGD | 100 | 0.884 | 33.78 | 7.62 |
| FGSM | 89.0 | 0.770 | 30.19 | 12.00 |
| FreqSAM | 99.4 | 0.978 | 40.24 | 3.18 |

## I   Comparison with Reported Baselines

Due to limited code availability for several recent publications, we do not reproduce some 2023–2025 methods as baselines in our evaluation. When discussing these works, we treat reported numbers as not directly comparable across differing experimental protocols, and we focus on qualitative and methodological differences.

## J   Runtime Analysis

Table 6 reports the average wall-clock time per image for generating adversarial examples on a single NVIDIA Tesla P100 GPU with $\epsilon = 8/255$ and 10 iterations.

Table 6: Average runtime (seconds/image) comparison of adversarial attacks on ResNet-50.

| Attack | Time (s/image) |
|---|---|
| FGSM | 0.006186 |
| PGD (10 steps) | 0.012775 |
| MI-FGSM | 0.013179 |
| DI-FGSM | 0.012911 |
| FreqSAM | 0.021330 |
| FreqSAM-Uniform | 0.028943 |
| FreqSAM-RandomMask | 0.028089 |

FGSM is the fastest due to its single-step update, while iterative methods (PGD, MI-FGSM, DI-FGSM) have similar runtimes around 0.013 s/image. FreqSAM incurs additional cost from saliency computation,

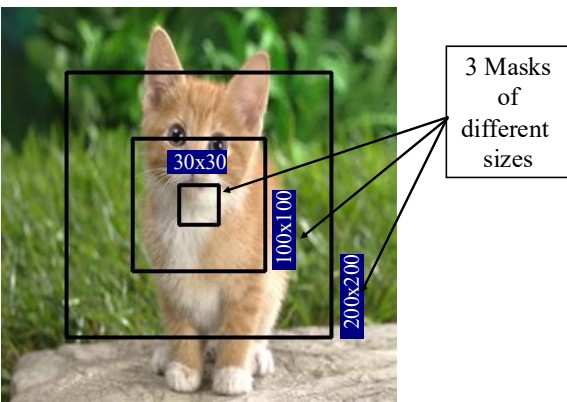

Figure 9: Effect of different mask sizes on a $299 \times 299$ image.

region selection, and FFT transforms. It requires 0.0213 s/image, around 1.6× slower than PGD. Other mask variants provide further speedups.

The additional computational cost of FreqSAM is modest and justified by its improved attack effectiveness and perceptual quality, making it a practical choice for generating high-quality adversarial examples.

## K    Effect of Mask Size

Figure 9 shows the effect of different mask sizes (30, 100, 200) on a $299 \times 299$ image, highlighting scale-dependent covering.

