# OpenReview forum: "FreqSAM: Saliency-Masked Frequency–Spatial Adversarial Attacks for Stealthy Examples"
_TMLR — Withdrawn by Authors_

### Review · Reviewer_vyuq · 2026-05-22

**Summary Of Contributions:**

The paper proposes FreqSAM, an adversarial attack that combines saliency-based spatial masking with FFT-based frequency-domain filtering. The goal is to generate adversarial examples with high attack success while improving perceptual quality according to metrics such as SSIM, PSNR, and L2. The method is evaluated on ImageNet across several architectures, transfer settings, and defense settings.

The combination of saliency localization and frequency-domain filtering is a reasonable idea. However, the main weakness is that the paper’s own results do not show FreqSAM to be a stronger attack. Its main advantage is improved distortion metrics, while older and simpler attacks often match or outperform it in ASR. The method is also significantly slower than standard iterative baselines.

**Audience:**

No

**Audience Explanation:**

In its current form, the findings do not provide enough new insight or practical value for the TMLR audience. The method combines known ingredients, namely saliency masking, iterative gradient updates, and FFT-based filtering, but the resulting attack is not stronger than established baselines and is slower than simpler PGD-style methods. The main finding is that localized perturbations improve distortion metrics such as SSIM, PSNR, and L2, which is not sufficiently surprising given that the perturbation is restricted to a smaller region. Since the paper does not show improved attack strength, does not evaluate full AutoAttack, and omits highly relevant defenses such as FrequencyLowCut Pooling by Grabinski et al. (2022), the contribution is currently too limited for TMLR.

**Broader Impact Concerns:**

The paper includes a broader impact statement. My main concern is that it reflects the same overclaiming as the rest of the paper. It presents FreqSAM as revealing critical vulnerabilities, but the empirical results do not establish a stronger attack than existing baselines. The broader impact discussion should be revised to reflect the actual contribution more accurately: FreqSAM improves selected perceptual distortion metrics for localized adversarial perturbations, but it does not establish a substantially stronger adversarial threat model.

**Claims And Evidence:**

No

**Claims Explanation:**

The paper’s claims are not convincingly supported by its own results. FreqSAM is repeatedly framed as a superior attack, but the reported tables show that its advantage is mainly in SSIM, PSNR, and L2, not in attack strength. Older and simpler attacks, such as PGD, APGD, and many more, already achieve higher ASR in several white-box settings, while SSA, APGDT, DI-FGSM, or VMI-FGSM are stronger in several defense or transfer settings. Table 6 also shows that FreqSAM is slower than PGD-style baselines, despite not providing a clear ASR advantage. The evaluation further misses key baselines, especially full AutoAttack and frequency/anti-aliasing defenses such as FrequencyLowCut Pooling by Grabinski et al. (2022), which might indeed render the attack completely ineffective since it is clear that the attack merely works by adding high-frequency noise in low-frequency regions. Therefore, the evidence supports at most a localized perceptual-quality attack, not the stronger claims made in the submission.

Reference:

Grabinski, J., Jung, S., Keuper, J., & Keuper, M. (2022). FrequencyLowCut Pooling - Plug and Play against Catastrophic Overfitting. In European Conference on Computer Vision, pp. 36-57.

**Requested Changes:**

- Substantially weaken the claims throughout the abstract, introduction, and conclusion. The current framing of FreqSAM as a superior attack is not supported by the results. The paper should state more narrowly that the method improves selected perceptual distortion metrics, not attack strength.
- Evaluate against the full AutoAttack suite, not only individual APGD/APGDT components.
- Include frequency-specific and anti-aliasing defenses. In particular, the paper should compare against FrequencyLowCut Pooling: Grabinski, J., Jung, S., Keuper, J., & Keuper, M. (2022). FrequencyLowCut Pooling - Plug and Play against Catastrophic Overfitting. ECCV. Related anti-aliasing pooling defenses, such as ASAP, should also be considered.
- Discuss Table 6 honestly. FreqSAM is slower than PGD, MI-FGSM, and DI-FGSM, while not being stronger in ASR. The current text does not justify this trade-off.
- Reframe the contribution as a localized perceptual-quality attack rather than a stronger adversarial attack.
- Clarify the transferability discussion, since the reported results do not support strong transferability claims for FreqSAM.

---

### Review · Reviewer_8HsP · 2026-06-16

**Summary Of Contributions:**

The paper proposes a frequency-spatial adversarial attack localized through saliency for image classification. The proposed method first identifies a salient region and then restricts adversarial perturbations to this region and shapes the perturbation in the Fourier domain using an FFT-based low-pass mask. The proposed method (termed FreqSAM) is evaluated  on ImageNet using several architectures, including ResNet-50, DenseNet-121, ViT-B/16, and Inception-v3. The reported results show that FreqSAM achieves high white-box attack success rates while substantially improving global perceptual metrics such as SSIM, PSNR, and (L_2) distortion compared to many standard full-image attacks.

The main strength of the paper is that it combines two intuitive and useful ideas: spatial localization through saliency and spectral shaping through frequency-domain filtering. The resulting method is simple, and appears to provide a favorable white-box effectiveness/perceptual-quality trade-off. The main weaknesses are that the novelty is incremental, the key components are not sufficiently isolated experimentally, and some comparisons may not be fully representative because FreqSAM perturbs only a local region while many baselines perturb the entire image.

**Audience:**

Yes

**Audience Explanation:**

The paper addresses adversarial robustness, which is an important topic for the TMLR audience. However, the way the paper presents results and ablations needs substantial improvement before the message is considered strong.

**Broader Impact Concerns:**

The paper already includes a Broader Impact Statement and acknowledges that adversarial attack methods can be misused to compromise deployed AI systems.

**Claims And Evidence:**

No

**Claims Explanation:**

The paper provides useful empirical evidence that FreqSAM achieves high attack success while improving global perceptual metrics. However, I do not think that all claims are supported by convincing evidence in the current version. Below, I am listing my concerns:

- First and foremost, as also acknowledged in the paper (introduction), the contribution is a combination of spatial attacks and frequency domain attacks. My main concern is that it is not clear what is the main idea of the paper and the ablation in Section 4.2.3 does not fully isolate the contribution of each component of the method. The ablation reposts comparisons against random masks, uniform variants, and Gaussian masks, but it does not sufficiently compare against more direct ablations such as: saliency-masked PGD without FFT filtering, FFT-filtered PGD without saliency masking, full-image low-pass FFT PGD, or saliency masking with different frequency bands. Without these comparisons, it is difficult to determine from which element the improvement is coming.
- A second concern is the fairness of the perceptual-quality comparison. Since FreqSAM perturbs only a localized region, global metrics such as PSNR, and SSIM will naturally be better than for attacks that perturb the full image. This does not necessarily demonstrate that the perturbation is less perceptible locally within the modified region. The paper should include local distortion and local perceptual metrics inside the mask, as well as stronger perceptual metrics in the main evaluation. Moreover L_2 distrortion and PSNR are highly correlated metrics (it is not clear why would the paper report both).
- The claims about transferability should also be toned down. The transferability results show that FreqSAM has moderate transferability, but it is clearly weaker than several strong transfer-oriented baselines such as DI-FGSM and SSA in many source-target settings.
- The defense evaluation is also not fully convincing. Against adversarially trained models, FreqSAM is competitive but not clearly superior to the strongest baselines. Against augmentation-based defenses, it is not fully clear whether the attack is adaptive to the defense, whether expectation over transformation is used, and whether the defenses constitute strong robustness baselines.

**Requested Changes:**

1. The paper should provide stronger ablations that isolate the role of each component (see comments above)
2. The paper should report comparisons with baselines in terms of perturbation support. Since FreqSAM perturbs a localized region while many baselines perturb the whole image, the paper should compare against localized versions of strong baselines.
3. Moreover, the paper should Include a stronger comparison with recent 2023--2025 attacks under the same evaluation protocol, rather than only discussing reported numbers from the original papers.
4. The authors should report local perceptual quality metrics inside the perturbed mask, not only global (L_2), SSIM, and PSNR. Global metrics are biased in favor of attacks that modify fewer pixels. Local (L_2), local PSNR/SSIM, LPIPS/DISTS in the main table, and possibly human-perceptual or detection-oriented metrics would make the stealth claims more convincing.
5. The transferability claims should be moderated. The results show that FreqSAM has moderate transferability but is generally weaker than strong transfer-oriented attacks such as DI-FGSM and SSA.
6. The experimental section should provide complete experimental details
7. The paper should also include a more precise discussion of novelty.
8. I would also suggest improving the writing and presentation; for example, some references are concatenated

---

### Review · Reviewer_zKTx · 2026-07-14

**Summary Of Contributions:**

The paper proposes a new adversarial attack (FreqSAM) that combines saliency-guided spatial localization with FFT-based spectral shaping to generate localized perturbations. The authors aim to improve the trade-off between attack effectiveness and perceptual quality by restricting perturbations to salient image regions while controlling their spectral characteristics. The proposed method is evaluated on ImageNet dataset across CNNs and ViTs, including experiments on transferability, adversarially trained models, and several perceptual quality metrics like SSIM, PSNR, etc.

**Audience:**

Yes

**Audience Explanation:**

Adversarial robustness remains an active area of ML research, and the paper investigates an interesting point in the design space between spatially localized and freq-domain perturbations. However, the contribution would be considerably stronger if the paper more clearly identified what new insight is obtained beyond combinding two existing ideas.

**Broader Impact Concerns:**

The paper already contains an appropriate broader impact discussion. I do not have additional ethical concerns beyond ensuring that the contribution is characterized proportionally to the empirical evidence presented.

**Claims And Evidence:**

No

**Claims Explanation:**

The experimental evaluation demonstrates that FreqSAM is capable of producing successful white-box attacks with favorable image-quality metrics; however, several of the broader conclusions rely on evidence that is either incomplete or not sufficiently diagnostic.

- First, the paper attributes the performance gains to the interaction between saliency localization and freq-domain shaping, but the current experiments do not convincingly disentangle the influence of these two mechanisms. The provided ablations mainly modify the masking strategy, but they do not isolate whether the observed gains originate primarily form restricting perturbations spatially, from the spectral filtering itself, or from their interaction. As a result, it is difficult to identify which component is actually responsible fro the reported improvements.

- Second, many of the reported advantages concern perceptual similarity rather than attack effectivness. Although a reasonable desgn objective, but the paper extrapolates these observations into broader claims regarding attack quality. In actuality, FreqSAM maintains competitive white-box performance while sacrificing some transferability relative to established transfer-oriented attacks. The discussion would therefore benefit from presenting this as a design trade-off rather than evidence of _overall superiority_.

- Third, the paper does not fully establish that the proposed attack exposes _qualitatively new vulnerabilities_. Most evaluations remain within conventional robustness benchmarks, and it is difficult to determine whether the observed behavior is specific to freq-aware perturbations or simply reflect the general effectiveness of iterative attacks under the chosen threat model. Further, the paper compares against APGD and APGDT individually, but these are only components of the AutoAttack benchmark. I don't see a direct comparison against AutoAttack, which has been a standard protocol for evaluating adversarial robustness.

**Requested Changes:**

- **Strengthen the empirical evidence for the proposed design:** The paper should more directly demonstrate which component(s) of FreqSAM are responsible for the observed gains. The current ablations mainly vary the masking strategy (random, uniform, Gaussian), but they do not clearly establish whether the improvements arise primarily from spatial localization, freq-domain filtering, or their combination.

---

- **Reposition the contributions:** Across Tables 2 and 3, FreqSAM achieves competitive white-box performance while consistently improving perceptual metrics, but it does not consistently outperform strong baselines on attack success against defended models or on cross-model transfer. I suggest authros to revise the abstract, introduction, and conclusion to emphasize the demonstrated strength of the method, namely, the trade-off between perceptual quality and attack effectiveness, rather than implying broad superiority across evaluation settings.

---

- **Strengthen the evaluation protocol:** The paper argues that freq-domain perturbations reveal weaknesses that are overlooked by conventional robustness evaluations, yet the current experiments do not directly test this hypothesis. For instance, the authors should validate their proposed method against _freq-aware_ defenses, for example, freq-based preprocessing, or low-pass filtering approaches, etc. Doing so would provide a more direct evidence that FreqSAM represents a _distinct_ freq-domain threat model. Further, since AutoAttack has become a standard protocol for evaluating adversarial robustness, including it would provide a stronger reference point for situating FreqSAM relative to established white-box attacks. Finally, I'd urge authors to report variance for the runs whose results are close to the reported baselines in order to establish significance.

---

### Note · Authors · 2026-07-16

I have read and agree with the venue's withdrawal policy on behalf of myself and my co-authors.